# Breaking Therapy Resistance: An Update on Oncolytic Newcastle Disease Virus for Improvements of Cancer Therapy

**DOI:** 10.3390/biomedicines7030066

**Published:** 2019-08-30

**Authors:** Volker Schirrmacher, Stefaan van Gool, Wilfried Stuecker

**Affiliations:** Immune-Oncological Center Cologne (IOZK), D-50674 Cologne, Germany

**Keywords:** NDV, viral oncolysis, immunogenic cell death, type I interferon, dendritic cells, active-specific immunotherapy, bispecific antibodies, gene therapy, checkpoint inhibition, T cell costimulation, RIG-I, IFNAR

## Abstract

Resistance to therapy is a major obstacle to cancer treatment. It may exist from the beginning, or it may develop during therapy. The review focusses on oncolytic Newcastle disease virus (NDV) as a biological agent with potential to break therapy resistance. This avian virus combines, upon inoculation into non-permissive hosts such as human, 12 described anti-neoplastic effects with 11 described immune stimulatory properties. Fifty years of clinical application of NDV give witness to the high safety profile of this biological agent. In 2015, an important milestone was achieved, namely the successful production of NDV according to Good Manufacturing Practice (GMP). Based on this, IOZK in Cologne, Germany, obtained a GMP certificate for the production of a dendritic cell vaccine loaded with tumor antigens from a lysate of patient-derived tumor cells together with immunological danger signals from NDV for intracutaneous application. This update includes single case reports and retrospective analyses from patients treated at IOZK. The review also presents future perspectives, including the concept of in situ vaccination and the combination of NDV or other oncolytic viruses with checkpoint inhibitors.

## 1. Introduction

Oncolytic viruses (OVs) provide a new promising way to treat cancer. Such biological agents replicate selectively in tumor cells and induce tumor-selective cell death (oncolysis). Oncolytic viral therapy has an initial phase in which the virus mediates direct oncolysis of tumor cells, followed by a second phase of post-oncolytic immune response. This post-oncolytic immune response is directed towards tumor-associated antigens (TAs) and is considered as a key factor for an efficient therapeutic activity [1]. 

OVs adapted to the human immune system, such as native *measles virus* and *herpes-simplex virus 1* (HSV-1) exert adverse effects on human dendritic cells (DCs). These negative effects include cell viability, maturation and expression of co-stimulatory molecules. *Measles virus, mumps virus,* and *respiratory syncytial virus* are *paramyxoviruses* from man and cause serious human diseases. 

Genetic engineering enabled to develop from all the mentioned viruses recombinant OV strains without pathogenicity. Reverse genetics engineering has allowed development from negative strand RNA viruses recombinant OV strains with additional transgenes [2]. 

A review from 2018 on oncolytic viro-immunotherapy of hematologic and solid tumors lists ten virus families from which new recombinant oncolytic strains have been generated: *Adenoviruses, flaviviruses, herpesviruses, orthomyxoviruses, paramyxoviruses, picornaviruses, poxviruses, reoviruses, rhabdoviruses, and togaviruses*. A plethora of trials are mentioned which have been initiated to assess the safety and efficacy of these OVs [3]. 

This review deals with a native OV from birds, *Newcastle disease virus* (NDV). This paramyxovirus is not adapted to the human immune system. Birds are permissive hosts of this virus, while cells from mammals, including man, are non-permissive. Since NDV has neither adverse effects on human cells nor any pathology, it can be used as a native OV in cancer patients. The safety profile for NDV includes lack of gene exchange via recombination, lack of interaction with host cell DNA, virus replication independent of cell proliferation and low side effects in cancer patients. 

Newcastle disease is a major obstacle in poultry industry worldwide [4]. Certain strains of NDV have been developed to be used for preventive vaccination of chickens for more than 60 years [5]. In the 1960s, the phenomenon of viral oncolysis was discovered and a search began for a type of virus most suitable for clinical application in cancer patients. 1965, William A. Cassel reported about NDV as an antineoplastic agent in man [6]. Since then, NDV has been applied to cancer patients in the USA and in Europe [4,5]. Meanwhile, new regulations require a high-quality standard for NDV production as prerequisite for clinical application. 

Findings from recent years have shown that NDV has the potential to break cancer therapy resistance. This review aims at updating information concerning NDV with regard to basics and application in cancer patients.

## 2. Basic Information

### 2.1. Evolution and Taxonomy of NDV

Mammals developed about 200 million years ago while a majority of bird species developed only about 66 million years ago [7]. Bird viruses thus had a relatively shorter time to adapt to the immune system of their hosts than viruses of mammals. Multicellular organisms, like birds and mammals can respond to virus infection, in particular by a type I interferon response (see below). As an avian virus, NDV has evolved viral immune escape mechanisms in birds. These interfere with the type I interferon mediated host response. Importantly, this viral escape mechanism is species specific and does not apply to non-permissive hosts.

NDV is an avian paramyxovirus type 1 (APMV-1). Such viruses have a negative sense single-stranded RNA (−ssRNA) as genome. Some strains show in non-permissive hosts a natural oncotropism (i.e., tumor selective viral replication), oncolytic potential and immune stimulatory properties. 

The phylogenetic classification system of NDV has recently been updated [8]. NDV strains are classified according to their pathotypes and virulence as either lentogenic (low), mesogenic (medium) or velogenic (high). Velogenic strains are highly infectious in birds and are distinguished as viscerotropic or neurotropic pathotypes.

### 2.2. Molecular Biology of NDV

Genome sequences for many strains of NDV are available on the web at www.ncbi.nlm.nih.gov. All genome sizes of NDV obey to the rule of six which is characteristic for APMV-1 [9]. The genomic RNA contains a 3′-extragenic region known as leader and a 5′-extragenic region known as trailer. These are regions for control of virus transcription and replication and also for encapsidation of newly synthesized RNAs into virus particles. Leader and trailer flank the six genes (3′-N-P/V-M-F-HN-L-5′) of the viral genome. The genes code for nucleoprotein (NP), phosphoprotein (P), matrix protein (M), fusion protein (F), hemagglutinin-neuraminidase protein (HN) and large protein (L). The genome has a large capacity (>5 kb) for the incorporation of transgenes [4]. 

Infection of cells by NDV involves as a first step binding of the virus to the host cell’s surface via the cell-binding domain of the HN molecule [10]. α2,6-linked sialic acid was demonstrated to serve as high-affinity receptor for HN and for binding of NDV to cells [11]. Cell surface binding of the virus is followed by activation of the fusion protein F. A concerted action of HN and F leads to fusion of the viral and the host cell membrane so that the viral genome can enter the cytoplasm of the host cell. NDV can also utilize a clathrin-mediated endocytosis route and macropinocytosis as alternative endocytic pathway to enter cells [12]. In the cytoplasm, the −ssRNA is transcribed into messenger RNA. The viral mRNAs remain polyadenylated and carry 5′-phosphate groups (5′-PPP) in contrast to mammalian mRNAs which are either capped or contain base modifications. The viral mRNAs are then translated into the corresponding viral proteins.

The second step relates to viral replication using a newly produced nucleocapsid as positive-strand “anti-genome” [13]. The viral nucleocapsid consists of a single species of viral RNA (15,186 to 15,198 nucleotides in length) and replicon complex proteins (NP, P, L). Once produced, the nucleocapsid as “antigenome” is used as template for viral replication. The viral proteins P and L associate to form the enzyme RNA-dependent RNA polymerase, which is essential for viral replication. Viral replication can produce 1000 to 10,000 copies per cell. The newly produced viral genomes become encapsulated at the plasma membrane. A membrane budding process then enables the release of new virus particles.

Infection of normal cells from non-permissive hosts (e.g., mouse or man) usually does not proceed to the second step [14]. In NDV-infected non-activated or activated T cells, intracellular viral RNA (− and + strands and double-stranded (ds) RNA) can be detected within 1 to 12 h [14]. The viral replication cycle is prevented by *interferon-induced genes* (ISGs) such as *myxovirus (influenza*) resistance A (MxA), 2′-5′-oligoadenylate synthetase/ribonuclease L (OAS) and dsRNA-activated protein kinase R (PKR) (13). Mammalian and avian cells use OAS/RNase L to degrade cellular and viral RNA and retinoic acid inducible protein I (RIG-I) to enhance interferon induction as the first line of defense against viral infection. Viruses have developed diverse strategies to escape antiviral effects of OAS. NS1 protein of *Influenza virus A* acts upstream of its pathway while other viral proteins such as *Theiler’s virus* L protein act downstream. 

NDV was recently reported to be able to establish persistent infection in tumor cells in vitro. There was a contribution of the cleavage site in F and of a second sialic acid binding site of HN. Persistent infection avoids oncolysis and immunogenic cell death (ICD) (see below) and facilitates viral spread from cell to cell as a potential mechanism to escape host antiviral responses [15].

OVs such as naturally occurring and attenuated NDV can be used alone or in combination with cancer vaccines for human cancer therapy [16]. OVs can also be used as vectors for gene therapy of cancer. Basic research in molecular biology and virology allowed to elucidate paramyxovirus replication and pathogenesis. Such studies were primarily performed with the rodent paramyxovirus *sendai virus*. Reverse genetics is a method by which infectious negative strand RNA viruses can be generated entirely from cloned DNA (c-DNA) [3,17,18,19]. This molecular engineering technology has allowed to develop recombinant oncolytic paramyxovirus strains with additional transgenes.

Harnessing OV-mediated anti-tumor immunity by therapeutic transgenes, pharmacological agents or bispecific antibodies has been the topic of a special issue of Frontiers in Oncology [1].

### 2.3. The Type I Interferon Response in Birds and Its Inhibition

Type I interferon (IFN-I) responses serve as a first line of host defense against microbe invasion. Type I interferons can induce tumor cell apoptosis and anti-angiogenesis via signaling through the common type I interferon alpha receptor (IFNAR). IFN-I can also exert direct effects on cells of the immune system [20].

It is of great interest to compare the interferon response to NDV in permissive cells from birds as natural hosts to the response in cells from non-permissive hosts including humans. Infection of chickens with virulent NDV is associated with severe pathology, morbidity and mortality. Recently, immune responses were described of mature chicken bone-marrow-derived dendritic cells (BM-DC) upon infection with NDV strains of differing pathogenicity. Gene expression profiling revealed increased expression of *melanoma differentiation associated gene 5* (MDA-5), of the helicase LGP2, the Toll-like receptors (TLR) 3 and 7, of type I interferon (IFN-α, IFN-β), IFN-γ, of interleukins (Ils) IL-1β, IL-6, IL-10, IL 12, IL-18, of chemokine ligand CCL5, and of major histocompatibility molecules MHC-I and MHC-II. Velogenic NDV showed a stronger replication capacity in BM-DCs than lentogenic NDV [21]. 

Of importance for the pathogenicity of NDV in birds is the capacity of certain viral products to antagonize the interferon response. In avian cells, NDV has developed a frameshift variant of the viral phosphoprotein P to escape type I IFN mediated anti-viral responses [22]. The incorporation of two G nucleotides at the RNA editing site of the P gene results in the frameshift variant V protein. Recently it was demonstrated that the interferon antagonistic activities of the V proteins of NDV correlated with their virulence [22]. The V protein interacts specifically with bird proteins [22]. It thereby can inhibit IFN signaling by targeting signal transducers and activators of transcription 1 (STAT1) for degradation. It can also interact with MDA-5, leading to the inhibition of interferon regulatory factor (IRF)-*3* activation and IFN-β induction [23]. 

Upon virus infection, MDA-5 (or RIG-I in humans) recognizes ds viral RNA with 5′-triphosphate as being foreign and distinct from self-RNA. This initiates a strong type I interferon response [24,25]. Mitochondrial antiviral-signaling protein (MAVS) is an essential adaptor protein in RIG-I mediated antiviral innate immunity. Recently, a MAVS gene from goose (goMAVS) was identified. This bird derived MAVS mediated the activation of the type I interferon pathway in a species-specific manner [26]. Cells from birds can activate [26] or inhibit [23,27] a type I interferon response quite similar to cells from mammals. However, decisive protein–protein interactions in the interferon signaling pathway are bird-specific.

### 2.4. The Type I Interferon Response in Mammalian Cells and Its Inhibition

In mammalian cells, NDV induces a strong interferon response which involves an early and a late phase. This leads to inhibition of virus replication. The early phase response is initiated by cytoplasmic RIG-I-like receptors (RLR) upon recognition of foreign viral RNA while the late phase is initiated by membrane-bound IFNAR upon recognition of their respective ligands [20,27]. RLR is a family of cytoplasmic RNA helicases that includes RIG-I and MDA-5. These RNA sensors signal through the mitochondrial adaptor MAVS, recruiting kinases to activate the nuclear transcription factors NF-κB and IRF-3, and induce the transcription of IFN-I and proinflammatory cytokines.

In RIG-I-deficient fibroblasts, cytokine production is abrogated in response to NDV, *sendai virus, vesicular stomatitis virus*, *hepatitis C virus* or *influenza A and B virus*, thus demonstrating the importance of this foreign RNA sensing cytoplasmic receptor [28]. 

Other pathogen recognition receptors (PRR) are TLRs. These consist of more than 10 members and are expressed on the cell surface membrane or on endosomes. TLRs appear to be required to induce an interferon response in plasmacytoid dendritic cells (pDC) [29], while RLRs are critical to sense NDV by conventional myeloid DCs (mDC), macrophages, and fibroblasts. Details about the interferon signaling cascades and their regulation have been described [30,31]. Cytosolic nonself DNA recognition via the cyclic GMP-AMP synthase (cGAS) pathway also leads to IFN-I induction [32]. Upon sensing DNA viruses, RNA polymerase III is reverse transcribed to produce short RNAs, which are recognized by RIG-I. Latest updates on RIG-I also reveal long noncoding RNAs (LncRNAs) as being involved in regulation of RLR pathways. Antiviral IFN-I signaling is also regulated by covalent protein modification via ligation of small ubiquitin-like modifier (sumoylation) [32]. Not only foreign nucleic acids but also “unmasked”, misprocessed, or mislocalized host-derived RNA or DNA molecules can be recognized by RLRs or cGAS thus leading to proinflammatory or autoimmune disease [32].

During the late phase (8–18 h) of the IFN response [25], the induced type I IFN molecules (IFN-α and -β) secreted during the early phase, interact with the cell surface expressed IFNAR. These cytokine receptors are expressed by cells of all lineages, but not on mature erythrocytes. Nearly 20 four-helix bundle cytokines exist in humans and mice to interact as ligands with IFNAR. IFNAR consist of two chains, R1 and R2. The cytoplasmic domains of these chains are physically associated with Janus-family tyrosine kinases (JAKs) and Tyrosine-kinase (Tyk)-2. Ligand–receptor interaction initiates an amplification loop of the IFN response, which involves signaling via STAT proteins and IRF-7 [33]. 

Human mDCs were infected by NDV to study an uninhibited cellular response to virus infection [34]. The new approach integrated genome-wide expression kinetics and time-dependent promoter analysis. Interestingly, the anti-viral cell-state transition during the first 18 h post-infection could be explained by a single convergent regulatory network. A network of 24 transcription factors was predicted to regulate 779 of the 1351 up-regulated genes. It was concluded that the proposed network is effective in changing the cells underlying biology. The timing of this step-wise transcriptional signal propagation appeared as highly conserved. The effect of NDV on human DCs was also analyzed by the release of cytokines important for Th1 or Th2 polarization. Human monocyte-derived DCs were found to become polarized towards DC1 [35]. Signaling through RIG-I and IFNAR was found in further studies to be of importance for immune activation of DCs and other immune cells by NDV in mouse or human [25].

In contrast, signaling through RIG-I and IFNAR is successfully antagonized by respective inhibitory proteins from *ebola virus* (EBOV). EBOV, belonging to the family of *filoviridae*, was first discovered in 1976. It infects cells from primates (*gorilla, shimpanzee, human*). Primary target cells are macrophages and DCs. Zoonotic transmission of EBOV to humans causes severe and often lethal hemorrhagic fever. Disease characteristics are a systemic inflammatory response syndrome (SIRS), disseminated intra-vascular coagulation, systemic hemorrhage and multiple organ failure. Two of the 8 viral proteins of EBOV are involved in immunosuppression. In multiple ways they prevent type I interferon signaling. VP35 antagonizes the early phase of the interferon response. VP24 is an antagonist of the late phase. The molecular details of this virus-inhibited cellular response to infection have been described elsewhere [36]. Recently, a novel EBOV glycoprotein (GP) modified recombinant NDV (rNDV) was constructed as potential vaccine. The EBOV GP was expressed at a high level. Upon immunization, guinea pigs developed high levels of neutralizing GP-specific IgG and IgA antibodies [37].

## 3. Anti-Neoplastic and Immune-Stimulating Effects

### 3.1. Virus Infection of Tumor Cells from Non-Permissive Hosts: Tumor-Selective Virus Replication

While infection of normal cells from mouse or human usually does not proceed to the second step of virus replication, in the majority of all tested murine or human tumor cells, NDV infection proceeded to the second step thus allowing tumor selective replication. Expression of RIG-I, IFNAR, IRF-3, IFN-β, and IRF-7 was reported to determine about resistance or susceptibility of murine cells (macrophages and macrophage-derived tumor cells) to infection by NDV [38]. The relative importance of these genes and their products was revealed by determining susceptibility to NDV infection of cells from respective gene knock-out mice [39]. In tumor cells, mutations in such genes impair the effectivity of the interferon response system. This allows for un-inhibited growth and relative resistance to apoptosis.

NDV was reported to replicate up to 10,000-fold faster in human cancer cells than in most normal human cells [39,40]. In cases of tumors with relative resistance to NDV virus replication, receptors such as RIG-I could be targeted to downregulate their activity. Targeting RIG-I-like helicases in pancreatic cancer was reported to induce ICD [41].

The molecular targets for NDV infection of neoplastically transformed cells could be identified. One study [42] made use of a skin carcinogenesis model consisting of a human keratinocyte non-tumorigenic cell line (HaCaT) and its multistage tumorigenic derivatives. A siRNA screening approach served to identify virus-sensitizing genes. The small Rho GTPase Rac1 was found to be essential both for anchorage-independent growth of the cells and for NDV replication. Rho GTPases belong to the branch of small GTPases of the Ras superfamily of oncogenes. Identification of point mutations in the Rho GTPases Rac1, RhoA, and Cdc42 in human tumors supports the new paradigm that Rho GTPases serve as oncogenes in several human cancers. NDV is thus likely the first OV with an identified oncogenic target molecule.

It was reported that NDV interacts with Rac1 (i) upon viral entry, (ii) during syncytium induction and (iii) upon actin reorganization [43]. Since Rac1 is also important for glioblastoma multiforme (GBM) cell migration and invasion, it has been suggested to use NDV for targeted therapy against GBM [44]. Interestingly, Rac1 signaling, actin rearrangement and plasma membrane ruffling have also been described during clathrin-mediated endocytosis of NDV [12]. 

Recently, differences have been described in responses of 11 human pancreatic adenocarcinoma (HPACs) cell lines to infection by oncolytic NDV [44]. While all cell lines were susceptible to NDV, there existed differences in replication kinetics and cytotoxic effects. Not all HPACs had functional defects in the interferon pathway [45]. In vivo studies revealed NK cell activation by NDV in HPACs and importance of an adaptive T cell mediated immune response [46].

Further updates of NDV studies relate to the role of nuclear localization of paramyxovirus proteins and their importance for virus life cycle, including the regulation of viral replication and the evasion of host immunity. Nuclear localization of the M protein was demonstrated to increase viral RNA synthesis and transcription efficiency in chicken fibroblasts and to promote the cytopathogenicity of NDV [47].

NDV virus replication in monkey kidney-derived Vero cells was described recently to be regulated also by sarco/endoplasmic reticulum calcium ATPase (SERCA). This is a membrane-bound cytosolic enzyme which regulates the uptake of calcium into the sarco/endoplasmic reticulum [48]. 

Other recent findings of relevance for NDV replication in tumor cells relate to the formation of syncytia, to autophagy and to viral exosomes. Co-transfection of cDNA from HN and F of virulent NDV induced syncytium formation and triggered complete autophagy [49]. This was mediated through the activation of the AMPK-mTORC1-ULK1 pathway [50]. MTOR is a protein kinase, ULK1 is an autophagy-initiating protein kinase and AMPK is an energy-sensing AMP-activated protein kinase [50]. The AMPK-MTORC1-ULK1 network plays a role in autophagy and in maintaining cellular energy and nutrient homeostasis. 

To investigate signal transduction pathways derived from NDV infection, human MCF-7 breast cancer cells were infected by NDV. Protein expression was analyzed by two dimensional western blots and by MALDI mass spectrometry and compared to uninfected cells. A major difference was found in the expression of a small Heat-Shock Protein (HSP27) which became phosphorylated [51]. HSP27 is a redox-sensitive molecular chaperone [52]. HSP27 and High Mobility Group Box 1 (HMGB1) play an important role in autophagy [53]. Autophagy is a lysosome-dependent cellular degradation program that responds to cellular stresses and is evolutionary well-conserved to maintain cellular homeostasis [54]. 

Exosomes are vesicles encapsulating RNA, DNA, and proteins for intercellular communication. To obtain NDV-related exosomes (NDV-Ex), a protein A/G-correlated method was used. These exosomes promoted NDV propagation. Viral NP proteins could be transferred to cells via NDV-Ex. They exhibited viral replication-promoting and cytokine-suppressing abilities [55]. NDV exosomes were reported also to carry microRNAs into neighboring cells. Three miRNA candidates embraced by exosomes were associated with enhancing NDV-induced cytopathic effects in HeLa cells. These miRNAs could suppress IFN-β gene expression. The involvement of NDV-employed exosomes in tumor cells was suggested to result in inhibition of the IFN pathway and promotion of viral infection [56]. 

### 3.2. Mechanism of Tumor-Selective NDV-Mediated Oncolysis and Immunogenic Cell Death

ICD is defined by endoplasmic reticulum (ER) stress response [46], generation of reactive oxygen species (ROS), emission of damage-associated molecular patterns (DAMPs) and induction of antitumor immunity. NDV mediates its oncolytic effect by intrinsic and extrinsic caspase-dependent pathways of cell death [57]. The activation of the NF-κB pathway and subsequent upregulation of TNF-related apoptosis-inducing ligand (TRAIL) initiates extrinsic apoptosis, leading to activation of caspase 8 and cleavage of Bid into tBid. tBid transmits the extrinsic apoptotic signals to mitochondria [58]. This causes opening of mitochondrial permeability transition pores, loss of mitochondrial membrane potential, and release of cytochrome C, leading to activation of the intrinsic apoptosis process with cleavage of caspase 9 [59,60]. p38 Mitogen Activated Protein Kinase (MAPK) pathway and ER stress pathways also play important roles in NDV-mediated oncolysis [61].

Progeny of lytic strains of NDV contain active F protein, whereas the progeny of non-lytic strains contain inactive F protein. Lentogenic strains behave as non-lytic virus, whereas mesogenic and velogenic strains behave as lytic strains [62]. Not only lytic but also nonlytic strains of NDV can kill cancer cells. 

Virus infection interferes with cancer cell metabolism: The competition between virus and host cell for the protein synthesis machinery causes an ER stress response. A stressful cellular state is coupled to the activation of the PERK/eIF2α/ATF4/CHOP pathway of the unfolded protein response (UPR). PKR, an interferon stimulated gene, upon detection of nonself dsRNA, coupled with eIF2α phosphorylation, induce stress granule (SG) formation. Critical antiviral effectors such as RLRs, OAS, DHX36, and RNaseL, are recruited into SG. Close proximity of RLRs with viral dsRNA within SG facilitates a more efficient detection, and robust signaling [28]. This is associated with phosphorylation of PERK and eIF2α [14], which then shuts off protein synthesis. 

FACS analysis of MCF7 human breast cancer cells revealed immunogenic changes upon infection by lentogenic NDV (strain *Ulster*). This involved (i) cell surface expression of viral HN and F molecules, (ii) upregulation of HLA and cell adhesion molecules (CAM), (iii) induction of interferons and chemokines, and (iv) induction of apoptosis [63]. Tumor cell surface expressed HN was demonstrated in another study to induce the activation of NK cells [64]. Glioma cells, upon infection by NDV, revealed the upregulation of calreticulin (Ecto-CRT) and of heat-shock proteins (HSPs) [65]. These events can be termed immunogenic apoptosis. NDV also induces in tumor cells necrosis with the release of cytokines (IFN-I, TNFα) and chemokines (RANTES, IP-10). The whole process is called immunogenic cell death (ICD) and involves also the recognition of Pathogen-associated molecular patterns (PAMPs) (e.g., viral RNA, viral protein HN) and the release of DAMPs (e.g., High Mobility Group Box 1 (HMGB1) and ATP) [66]. 

Identified mechanisms of selectivity of NDV for tumor cell oncolysis in non-permissive hosts include the following: (i) defects of tumor cells in activation of anti-viral signaling pathways (via RIG-I), (ii) defects of tumor cells in type I IFN signaling pathways (via IFNAR), (iii) defects of tumor cells in apoptotic pathways [61].

The oncolytic potential of R_2_B Mukteshwar vaccine strain of NDV has been evaluated in a human colon cancer cell line xenografted in nude mice. 10^7^ plaque forming units were applied via the intratumoral (i.t.) or intravenous (i.v.) route. There was a 43% and 53% tumor growth inhibition in case of the i.t. route with regard to absolute and relative tumor volume, respectively, and 40% and 16% inhibition, respectively, via the i.v. route. The virus survived in 2/2 mice until day 10 and in 3/6 mice by day 19, with both routes of administration. The test NDV strain was found to be safe and showed oncolytic activity against the SW-620 colon cancer cell line in mice [67].

Recently, a Malaysian field outbreak isolate, NDV strain AF2240, has been characterized as having high oncolytic activity. NDV-mediated apoptosis involved Bax protein recruitment as well as death receptor engagement. Tumor selectivity of apoptosis was proposed to be due to a p38MAPK/NF-κB/IκBα pathway [68]. Regression of solid breast tumors in mice by this NDV strain was associated with production of apoptosis-related cytokines (IL-6, IFN-γ, MCP-1, IL-10, IL-12p70, and TNFα) [69].

### 3.3. Post-Oncolytic Immunity and Direct Activation of Immune Cells by NDV

In an orthotopic GBM model in immunocompetent mice it was demonstrated that NDV-mediated virotherapy induces long-term survival and tumor-specific immune memory [65]. In vivo, NDV-treated tumors became infiltrated with T cells secreting IFN-γ. Myeloid derived suppressor cells (MDSCs) showed reduced accumulation. When using immunodeficient Rag2 (−/−) mice or mice depleted of CD8+ T cells, no such therapeutic effects could be seen. The therapeutic effects relied on the induction of ICD in the GL261 glioblastoma tumor cells which led to induction of a post-oncolytic adaptive anti-tumor immune response [65]. 

Another recent study evaluated ultra-microscopic changes and proliferation of apoptotic GBM cells induced by the velogenic malaysian NDV strain AF2240. Analysis of the cellular DNA content showed that there was a loss of treated cells in all cell cycle phases (G1, S and G2/M) accompanied with increases in the sub-G1 region (apoptosis peak) [70].

Apart from induction of ICD and post-oncolytic anti-tumor immunity, NDV can also activate cells from the immune system directly.
NK cells: The viral surface protein HN of NDV is recognized by activatory NK cell receptors (NCR1) NKp46 and NKp44 of murine NK cells which transmit cytotoxicity inducing signals [64]. These two receptors are associated with the signaling chains CD3ζ (to NKp46) and DAP12 (to NKp44), respectively, which contain immunoreceptor tyrosine-based activation motif (ITAM) domains that facilitate signal transduction.Monocytes and macrophages: Upon contact with NDV, monocytes become activated to produce tumoricidal activity through TRAIL [71]. NDV activated macrophages exert anti-tumor activity not only in vitro [71] but also in vivo [72]. NDV-infected macrophages produce nitric oxide (NO) via activation of NFκB [73]. Caspase activation was required for NO-mediated, CD95-dependent and independent apoptosis in human neoplastic lymphoid cells [74]. The mechanism of NO-induced apoptosis in tumor cells has been reviewed [75].Dendritic cells: The effect of NDV on human DCs has already been summarized above. The antiviral response occured within 18 h, was highly reproducible and regulated by a choreographed cascade of transcription factors [34]. This response programmed the DCs towards DC1 [35].

When exposed to microbial stimuli such as NDV, DCs activate NF-κB. This induces a module for expression of pro-inflammatory cytokines together with a module for antigen presentation. These promote the generation of CD4+ T helper 1 responses and of CD8+ effector T cells. Such transcriptional determinants of immunogenic states during DC maturation where recently compared to those of tolerogenic DCs. The transcription factor interferon regulatory factor 4 (IRF-4) was found to promote, in the absence of microbial products, the generation of regulatory T cells (T regs). It was concluded that DCs can modulate their transcriptionally regulated modules. A core antigen presentation module can be directed by regulatory modules to prime tolerogenic or immunogenic T cells to promote either tolerance or immunity [76]. Through DC stimulation, IFN-α mediates a link between innate and adaptive immunity [31]. IFN-α influences antigen processing by modulating proteasome activity and boosts the epitope cross-presentation ability of monocyte-derived DCs [77]. IFN-α promotes intracellular antigen persistence, regulates intracellular pH and sustains the survival of antigen-presenting DCs by selectively upregulating anti-apoptotic genes [78]. DCs loaded with the lysate of tumor cells infected with NDV (viral oncolysate, VOL) trigger potent anti-tumor immunity by promoting the secretion of IFN-γ and IL-2 from T cells [79]. DCs loaded with VOL showed increased levels of CD80, CD86 and CD83 compared to DCs loaded with tumor cell lysate without NDV [79]. Furthermore, DCs pulsed with VOL stimulated autologous memory T cells from the bone marrow of breast cancer patients thereby releasing increased titers of INF-α and IL-15 [51]. Based on these findings, IOZK developed the advanced therapy medicinal product IO-VAC^R^ [80]. It is a next-generation DC vaccine that is integrated into the current cancer immunotherapy landscape [81].
4.T cells: HN molecules at the surface of infected tumor cells introduce new cell adhesive strength for interaction with lymphocytes [82,83]. Modification of tumor cells by a low dose of NDV introduced T cell costimulatory activity and augmented in vivo tumor-specific T cell response as a result of CD4+ and CD8+ immune T cell cooperation [84]. HN was demonstrated to augment peptide-specific CD8+ T cell responses [85]. Secondary tumor-specific cytotoxic T cell (CTL) responses in vitro were augmented when stimulator tumor cells were infected by NDV [86]. Such potentiation of CTL activity in vitro was mediated via induction of IFN-α, β (IFN-I) [87]. IFN-I plays an important role for the generation of CTLs not only in vitro but also in vivo. It was hardly possible to induce a CTL response in mice pretreated with an IFN-I neutralizing antiserum [87]. NDV infection of human melanoma cells was reported to break tolerance of a melanoma-specific CD4+ T helper cell line. This T-cell costimulatory activity was independent of B7-1/B7-2 [88].

Further studies about HN and the interferon response revealed the following: (i) Transfected HN but not F could induce IFN-α and TRAIL in human blood mononuclear cells [89], (ii) this response was mediated via HN lectin–cell interaction [90], (iii) there exist two ways to induce innate immune responses in human PBMCs: paracrine stimulation of IFN-α responses by viral protein (HN) or viral RNA [91], (iv) the HN gene was found to be a powerful molecular adjuvant for DNA anti-tumor vaccination [92].

### 3.4. Roles of Virus Infection and IFN-I for Interactions between NK Cells, DCs, and T Cells

An NDV effect in vivo could be traced down to a positive effect on CD4+ and CD8+ T-T cell interaction [84]. CD4+ helper T cell interactions with B cells on one hand and CD8+ T cells on the other hand are known since a long time.

Recently, it was suggested that also DCs interact which each other. DC exosomes (DCex) were reported to cross-present TLR ligands and to activate bystander DCs (DC–DC cell interaction) [93]. 

NK cells and DCs can exchange bidirectional activating signals (NK–DC cell interaction) in a positive feedback, as follows: mDCs and natural IFN-I producing pDCs are involved in NK cell activation. Reciprocally, activated NK cells activate DCs. They induce the maturation of mDCs, enhance their ability to produce pro-inflammatory cytokines and polarize them towards DC1. Such DCs have an up to 100-fold enhanced ability to produce IL-12p70 in response to subsequent interaction with Th cells [94]. 

In vivo priming of NK cell responses to viral and bacterial pathogens requires the presence of CD11c (high) DCs. NK cell priming is dependent on the recognition of type I IFN signals from DCs. DC-derived IL-15 is also necessary for the priming of NK cells, revealing a striking homology to T lymphocytes [95]. 

NK-T cell interaction: T cells lacking IFNAR are highly susceptible to NK cell mediated killing because they express ligands to NK cell activating receptors. Since NDV induces type I IFN and activates NCR1 on NK cells [64], the possibility exists that such activated NK cells attack T cells. Interestingly, IFN-I was recently reported to protect T cells against NK cell attack [96].

DC-T cell interaction: DCs can efficiently present exogenous antigens by MHC class I molecules, a phenomenon called cross-presentation. DCs cross-presenting antigens can either induce an adaptive T cell response (cross-priming) or they can prevent such a response (cross-tolerance), depending on the context of antigen-presentation. While IFN-I facilitates cross-priming [97], TGF-β facilitates cross-tolerance [98]. DCs cross-presenting viral antigens derived from apoptotic debris of autologous macrophages infected by human cytomegalovirus can serve as a sensitive tool to detect CD8+ MTC responses from HCMV-positive patients after transplantation [99]. Similarly, DCs cross-presenting TAs from apoptotic debris of VOL can reactivate tumor-reactive CD8+ MTC responses [51].

Furthermore, it was reported that the activation of memory CD8+ T cells via antigen cross-presentation required the presence of IL-2 [100]. This soluble noncognate signal mediated a helper effect of cognate interactions between CD4+ T cells and DCs as well as between DCs and CD8+ T cells [100].

Bidirectional cell stimulation was also observed during cognate interactions of T cells and DCs from the bone marrow of breast cancer patients. Such interactions between cancer-reactive MTCs and TA-presenting DCs led to improved cell stimulation, cell survival and antitumor activity in vivo [101]. Table 1 contains a summary of the immune stimulatory properties of NDV.

## 4. Potential of NDV to Break Therapy Resistance and Anti-Viral Immunity

That oncolytic NDV has the potential to break resistance of cancer cells to a variety of therapies has been suggested in 2015 [102]. Because this is of great importance, this update, four years later, appears to be justified.
Breaking resistance to chemotherapy (CT) or radiotherapy (RT): These therapies require cells to be in a proliferating state. Non-proliferating cells such as cancer stem cells or dormant tumor cells are not affected by CT or conventional photon radiotherapy using X-rays and gamma rays. In contrast to cytostatic drugs and RT, oncolysis by NDV does not depend on cell proliferation. Since NDV replicates in the cytoplasm of cells, it is independent from DNA replication. The virus is capable to replicate in X-irradiated cells such as ATV-NDV vaccine cells [103]. There is thus a potential of oncolytic NDV to target cancer stem cells and dormant tumor cells. In addition, NDV has the potential to break drug-resistance of cancer cells: (i) NDV was reported to induce apoptosis in cisplatin resistant human lung adenocarcinoma cells in vitro and in vivo [104]. Apoptosis was induced via the caspase pathway, particularly involving caspase-9. (ii) In another study, treatment with autophagy modulators was found an effective strategy to augment the therapeutic activity of oncolytic NDV against drug-resistant lung cancers [105]. (iii) Multidrug resistance, particularly resistance to temozolomide (TMZ), is a challenge in the treatment of GBM. NDV was found to enhance the growth-inhibiting and proapoptotic effects of TMZ on GBM cells in vitro and in vivo [106].Breaking resistance to apoptosis: Oncolytic NDV was demonstrated to be capable to break resistance to apoptosis. It was reported to even have selectivity for apoptosis-resistant cells [107]. Utilizing a human non-small-cell lung cancer cell line (A549) overexpressing the antiapoptotic protein Bcl-xL, the authors showed significant enhancement of oncolytic activity and NDV replication in comparison to A549 cells not resistant to apoptosis. The increased oncolytic activity seen was secondary to enhanced viral replication and syncytium formation [107]. Another study was performed with apoptosis-resistant primary melanoma cells that overexpress the inhibitor of apoptosis protein Livin [108]. In that study, NDV-HUJ, a one-cycle replicating virus, was capable to overcome the resistance of this cell line to apoptosis. Under the robust apoptotic stimulation of NDV-HUJ, caspases could cleave Livin to create a truncated protein with proapoptotic activity. Both studies reported that the interferon system was not involved in this type of NDV-induced oncolysis [107,108].Breaking resistance to hypoxia: Solid tumor microenvironments contain regions of hypoxia, in which a transcription factor, hypoxia inducible factor (HIF) is active. It influences gene expression and contributes to the tumors radio-, and chemo-resistance. A velogenic NDV strain was applied to compare the oncolytic effect against a clear cell renal carcinoma line under normoxic and hypoxic conditions. It was found that hypoxia augmented oncolytic activity regardless of the cells HIF levels [109].Breaking resistance to TRAIL: Trail-resistant hepatocellular carcinoma-derived cell lines were found to be more susceptible to NDV-mediated oncolysis than TRAIL-sensitive cells. IFN-stimulated gene (ISG)-12a over-expression or silencing enhanced or reduced the cells TRAIL sensitivities [110].Breaking resistance to immune checkpoint blockade: Zamarin et al. reported in 2014 that localized (i.t.) oncolytic virotherapy with NDV in B16 mouse melanoma could break systemic tumor resistance to immune checkpoint blockade immunotherapy [111]. The therapeutic effect was associated with marked distant tumor infiltration with activated CD8+ and CD4+ effector but not with regulatory T cells, and was dependent on CD8+ T cells, NK cells, and type I interferon. In 2018, the same group explored the immunogenic potential of NDV in bladder cancer. Immunotherapy in bladder cancer with checkpoint inhibitory antibodies revealed only suboptimal response rates. NDV was able to infect human and mouse bladder cancer cells and induced ICD. This was associated with upregulation of MHC and PD-L1 and occured even in cell lines that had been resistant to NDV oncolysis. In vivo, in a bilateral flank test model, localized NDV therapy was able to potentiate checkpoint blockade therapy in both treated and distant tumors and even when employing lysis-resistant tumor cells [112]. The latter study is reminiscent of a study from 2007, in which a host mediated anti-tumor effect of oncolytic NDV was described after intra-hepatic locoregional virus application when using a luciferase-transfected NDV-lysis resistant murine colon carcinoma cell line [113].Breaking resistance to anti-viral immunity: Anti-viral immunity is considered as a major hurdle for effective therapeutic activity of oncolytic viruses [1]. Surprisingly, in a recent study from Ricca et al. it was reported that pre-existing immunity to oncolytic NDV potentiates rather than inhibits its immunotherapeutic efficacy [114]. This suggests for NDV a potential to break resistance to anti-viral immunity.

Table 2 contains a summary of the anti-neoplastic effects of NDV.

## 5. Comparison of NDV to other OVs 

For comparison we select recombinant gene modified oncolytic viruses which either are already approved for clinical application (HSV: T-VEC) or those with promising clinical data (e.g., adenoviruses, coxsackievirus). Only some information can be given relating to immunogenicity, type I IFN response, anti-neoplastic effects and to the question of breaking therapy resistances. More detailed information about viral therapy of cancer can be found in a book encompassing 22 chapters [115] and in a review about viral vectors in gene therapy [116].

*Herpes simplex virus* (HSV) is an enveloped double stranded neurotropic DNA virus that naturally occurs in humans. HSV-1 is often associated with common cold sore and herpes keratitis but it can cause life-threatening encephalitis if infection spreads to the brain. Through molecular or genetic dissection it was shown that selective editing of the γ_1_34.5 gene enabled viral replication in malignant cells, activation of IRF3 and subsequent induction of a type I IFN response. This looks similar to NDV which naturally occurs in birds. The vector HSV-1 ΔN146 precluded phosphorylation of translation initiation factor eIF2α, ensuring viral protein synthesis. In an aggressive breast carcinoma model in vivo, the virus profoundly reduced primary tumor growth and metastasis burden [117].

Anti-tumor immunity of oncolytic HSV can be enhanced by activation of plasmacytoid DCs (pDCs). Upon activation by TLR agonists or viruses, pDCs develop cytotoxic activities. They can turn cold tumors into hotspots by recruiting immune cells to the site of inflammation. HSV vectors may contribute to stimulation of memory-type adaptive immune responses through presentation of TAs [118].

The latest development in HSV technology is a fusion-enhanced oncolytic platform. Following deletion of the genes encoding ICP34.5 and ICP47 to provide tumor selectivity, the immunogenicity is enhanced through insertion of a gene encoding a truncated, constitutively highly fusogenic form of the envelop glycoprotein of gibbon ape leukemia virus (GALV-GP-R-). This leads to enhanced ICD in vitro with increased release of HMGB1 and ATP and increased levels of calreticulin on the cell surface. In a rat syngeneic tumor model, the fusion gene expression increased abscopal uninjected tumor responses [119].

With regard to T-VEC, a recent review summarizes its mechanism of action and its clinical efficacy and safety [120]. Results from only one study, the randomized, Phase III open-label OPTiM study, shall be mentioned. It evaluated T-VEC in comparison to GM-CSF in *n* = 436 patients with unresected stage IIIB, IIIC or IV melanoma. Median OS in the T-VEC treatment arm was 23.3 months compared to 18.9 months in the control arm. The difference of 4.4 months was significant. Treatment-related grade 3 or 4 adverse events (AEs) occurred in 11% in the T-VEC arm and in 5% in the GM-CSF arm. T-VEC related AEs were fatigue, chills, pyrexia, nausea, flu-like illness and injection-site pain [120]. 

*Adenoviruses.* These human DNA viruses are non-enveloped and harbor a linear double-stranded genome of about 30–40 kb. They are a frequent cause of upper respiratory tract infections and have also been associated with gastroenteritis and pneumonia in young children. The E3-19K protein inhibits expression at the cell surface of MHC-I molecules and of MHC-I chain related A and B proteins (MICA/MICB) thus protecting infected cells from T cell and NK cell attack [121]. 

Over the last decade much progress has been made in elucidating the adenoviral life cycle. One of the first clinical trials demonstrating antitumor efficacy in a specific cancer used a replication-conditional adenovirus. This virus, ONYX-015, is defective in the early regulator protein E1B which binds to and inactivates p53. In 1999, an 18-year-old patient with ornithine-cytosine transferase deficiency died as a direct consequence of adenoviral gene therapy. A replication defective adenovirus expressing the ornithine-cytokine transferase enzyme was administered through the hepatic artery. The viral titer used was too high and elicited a cytokine storm syndrome. This incident was the first death in 10 years of gene therapy clinical trials involving more than 3500 patients. Meanwhile, new standards have been introduced by the NIH and FDA to improve the quality and safety of clinical trials.

Meanwhile, genetic modifications of oncolytic adenovirus 5 (Ad5)-based vectors enable enhanced viral replication, oncolysis, and post-oncolytic immune responses [121]. The mechanism of oncolytic Ad5-mediated tumor suppression involves virus-induced activation of the autophagic machinery in tumor cells. Tumor cell infection by this virus induces autophagy and subsequent death of tumor cells rather than enhancing their survival [122]. One vector, Ad5/3-hTERT-E1A-hCD40L, has been modified in several ways: it contains a chimeric Ad5/3 capsid for enhanced tumor transduction, a human telomerase reverse transcriptase (hTERT) promoter for tumor selectivity, and human CD40L for increased costimulation. Data obtained from two syngeneic mouse tumor models revealed that adenovirus coding for CD40L mediated multiple antitumor effects including oncolysis, apoptosis, induction of T-cell responses, and upregulation of Th1 cytokines [123]. Another vector, Ad5/3-E2F-d24-hTNF-a-IRES-hIL-2, induced abscopal effects in non-injected tumors thus revealing systemic antitumor efficacy [124]. Type I IFN signaling was required for the induction of antigen-specific CD8+ T cell responses in the gut mucosa following intra-muscular adenovirus vector vaccination [125].

There have also been studies of combining adenovirotherapy with adoptive T-cell therapy for the treatment of cancer. Intratumoral adenovirus injection caused proinflammatory effects on the tumor microenvironment. This led to expression of costimulatory signals on CD11c+ antigen-presenting cells and subsequent activation of T cells, thus breaking the tumor-induced peripheral tolerance [126]. Recently, it was reported that a tumor targeting oncolytic adenovirus can improve therapeutic outcomes in chemotherapy resistant metastatic human breast carcinoma (P). It was suggested that upregulation of coxsackievirus-adenovirus receptor (CAR) induced selective vulnerability of chemotherapy-resistant tumors [127].

*Coxsackievirus.* This type of picornavirus is a human enterovirus with a single-stranded + RNA genome and without an envelope. The oncolytic coxsackievirus A21 (CVA21) has shown promise as a single agent in several clinical trials and is now tested in combination with immune checkpoint blockade. One study supports the development of CVA21 as an immunotherapeutic agent for the treatment of acute myeloid leukemia (AML) and of multiple myeloma (MM). CVA21 stimulated potent anti-tumor immune responses. These included cytokine-mediated bystander killing, enhanced NK cell-mediated cellular cytotoxicity and priming of tumor-specific CTLs [128]. CVA21 also induced immunogenic apoptosis in bladder cancer cell lines, as evidenced by ICD determinants calreticulin and release of HMGB1. Vaccination of mice with syngeneic MB49 bladder cancer cells undergoing CVA21 induced ICD induced in these mice the ability to reject MB49 tumors [129].

FOLFOX, a combination of leucovorin calcium, fluoruracil, and oxaliplatin, is the first-line chemotherapy for stage III and stage IV colorectal carcinoma. Patients with FOLFOX-resistant CRC have poor prognosis. Coxsachievirus A11 (CVA11) was found to display potent oncolytic activity in human CRC cells in vitro and in vivo. This virus was potently oncolytic against the oxaliplatin-sensitive Caco-2 cell line, but had little effect on the oxaliplatin-resistant line WiDr. However, oxaliplatin pretreatment sensitized oxaliplatin-resistant CRC cells to lysis by CVA11 infection in vitro and in vivo [130].

In conclusion, one can observe similarities between native NDV and recombinant OVs from different virus families which are in development for cancer therapy. The similarities relate to ICD, to importance of type I IFN signaling and to induction of post-oncolytic T cell mediated anti-tumor immunity. In an interesting additional study, apart from the above viruses also other OVs such as *measles virus, vaccinia virus*, or *vesicular stomatitis virus* were documented to be immunogenic. They were demonstrated to sensitize human tumor cells for NY-ESO-1 tumor antigen recognition by CD4+ effector T cells [131].

With regard to breaking cancer therapy resistances, NDV appears quite superior to the various OV vectors which are being developed by pharmaceutical companies. How nature has accomplished this superiority remains to be further elucidated. Also, in comparison to NDV, the elucidation of the molecular immunobiology of the other OVs is lacking behind. 

## 6. Over 50 Years of Clinical NDV Application

### 6.1. 1960s to 1970s: Post-Operative Treatment with Oncolysate Vaccines

As early as 1965, Cassel and Garret from Atlanta (GA, USA) reported on oncolytic NDV (strain *73 T*) as an antineoplastic agent [6]. Thereafter they observed the development of post-oncolytic immunity [132]. These authors were pioneers in developing NDV-based viral oncolysate vaccines for post-operative active-specific immunization of stage II malignant melanoma patients (two Phase II clinical studies involving 32 and 51 patients). A ten-year follow up of these 83 treated patients revealed that over 60% were alive and free of recurrent disease [133].

Later, a similar Phase II study with autologous NDV-modified tumor lysate vaccines has been conducted in Germany involving 208 patients with locally advanced renal cell carcinoma. Kirchner et al. concluded that the results demonstrated improved disease-free survival (DSF) in comparison with survival data published for similar patients who were treated by surgery alone [134].

### 6.2. 1990s to 2000s: Post-Operative Treatment with Autologous Live NDV-Modified Tumor Cell Vaccine (ATV-NDV)

To increase the immunogenicity of virus-modified tumor vaccines, a new concept was developed at the German Cancer Research Center (DKFZ) in Heidelberg, Germany. The idea was to develop an irradiated virus-infected autologous live cell tumor vaccine (ATV-NDV). The concept was tested with success in various metastatic animal tumor model systems [135,136]. After a series of tests with human tumor cell lines or with tumor cells from fresh operation specimens, human tumor cell modification by virus infection (NDV lentogenic strain *Ulster*) was found to be an efficient and safe way to produce a homologous human ATV-NDV cancer vaccine. This had pleiotropic immune stimulatory properties [137].

One clinical study evaluated the effect of vaccine quality parameters (i.e., cell number and cell viability) on the survival of primary operated breast cancer patients treated post-operatively with ATV-NDV [138]. Overall survival (OS), four years after surgery, was 96% for patients who received a high-quality vaccine (*n* = 32) compared with an OS of 68% for those who had received a low-quality vaccine (*n* = 31), a difference that was statistically significant [139].

A Phase II trial involved patients with locally advanced colorectal carcinoma (*n* = 57). Two years after surgery, OS for patients treated post-operatively with ATV-NDV was 98% compared to 74% for historical control subjects [138]. Using ATV-NDV vaccine obtained from cell culture, a further Phase II study was performed with patients suffering from glioblastoma multiforme. The median OS of vaccinated patients (*n* = 23) was 100 weeks versus 49 weeks in 87 non-vaccinated control subjects from the same clinic and the same time period (*p* < 0.001) [140].

A similar Phase II trial involved 20 patients with stage III and IV head and neck squamous cell carcinomas (HNSCC). The study demonstrated feasibility and safety of the ATV-NDV vaccine regimen and suggested prolongation of five-year survival [141].

Finally, a prospectively randomized Phase II/III trial (*n* = 50) was performed in stage IV colorectal carcinoma patients after resection for hepatic metastases. The objective was to investigate the efficiency of ATV-NDV as a tertiary prevention method. Evaluation was performed after an exceptionally long follow-up period of 9–10 years. While there was no significant difference between control and vaccinated rectal cancer, a significant benefit was seen in the colon cancer subgroup with regard to metastasis-free survival and overall survival: In the vaccinated arm, only 30.8% had died, while in the control arm 78.6% had died [142]. The trial provides evidence for the clinical value of the vaccine ATV-NDV. Its mechanism of function has been reviewed [143].

In 2003, a Phase III trial from China reported results from 310 patients suffering from Gastrointestinal Carcinoma (stage I-IV) which were treated with autologous tumor vaccine and NDV vaccine (strain *La Sota*). In comparison to 257 patients in which resection was performed without further vaccination, a significantly improved mean and median survival was observed for the vaccination group (7 years versus 4.46 years) [144]. 

### 6.3. 2000s: Systemic Application Studies with Oncolytic NDV

A selected case series study of four high grade GBM was performed in Hungary by Csatary and colleagues by i.v. application of an attenuated oncolytic veterinary vaccine strain, which was termed MTH-68/H. It was reported in 2004 on radiographically-documented responses and on long survival with improved symptomatology [145]. About ten years earlier, the same group had reported on a placebo-controlled Phase II clinical trial in which the virus was applied to patients (*n* = 26) with various advanced chemorefractory cancers. Interestingly, the virus had been applied at high doses via inhalation as a means to target lung metastases [146]. The high virus doses had been well tolerated. The 2-year survival rates had been 21% in the virus treatment group versus 0% in the placebo group. Although the study had not been randomized, the authors suggested a decrease in cancer-related symptoms and better survival [146].

Another case-series study in GBM was performed in Israel by Freeman, Zakay-Rones and colleagues with the lentogenic strain NDV-HUJ. The virus was administered i.v. using intra-patient dose-escalation followed by three cycles of 55 billion infectious units. Toxicity was minimal and a maximal tolerated dose was not reached. One patient experienced a complete response, although of only transient duration (about three months), while the others developed progressive disease [147].

The oncolytic NDV strain *PV701* was applied in the United States by Wellstat Biologics in patients with advanced cancers that were unresponsive to standard therapy. This strain had been extensively tested in vitro [39] and in human tumor xenotransplanted mice [40]. Seventy-nine late-stage cancer patients were given escalating doses of virus i.v. Doses of 12 × 10^9^ to 12 × 10^10^ infectious particles (plaque forming units/m^2^) were well tolerated [148]. 3 Phase I trials were undertaken, involving 113 patients, to evaluate effects, such as virus dose, schedule and i.v. infusion rate. Adverse events were flu-like, tumor-site-specific or those occurring during infusion. When patients were desensitized with a lower initial dose, the maximal tolerated dose (MTD) could be increased 10-fold. In 95 evaluable patients, there were 10 responses (six major and four minor) with progression-free survival ranging from 4 to 31 months [149]. 

Another interesting activity of NDV relates to the clinically important process of liver fibrosis. Activated hepatic stellate cells (HSCs) represent a crucial factor in the development of liver fibrosis and are involved in the development of hepatocellular carcinoma (HCC). NDV was reported to be able to repress the activation of human HSCs and to be capable, upon systemic application, to revert the development of hepatic fibrosis in mice [150]. 

### 6.4. 2010s: Combining Oncolytic Virus Modified Vaccines (ATV-NDV) with Costimulatory Bispecific Antibodies 

Most of the clinical trials performed with the vaccine ATV-NDV showed promising results with improvements in OS by about 30% of the treated patients. The remaining about 70% have to be considered as immunological non-responders. Since T cell anergy (non-responsiveness of TA-specific T cells) is a major problem in cancer patients and since this is often due to insufficient costimulation, a strategy was developed, as early as 1999, to augment T cell costimulatory signals in the vaccine ATV-NDV [151].

The strategy consisted of adding NDV-specific single chain antibodies with dual specificity (bispecific scFv antibodies, bsAb) to the vaccine ATV-NDV. In comparison to other bispecific antibodies, such as bispecific T cell engagers (BITES), which target TAs, the above bsAbs target viral antigens (VAs). They can thus be applied to any type of tumor cell modified by NDV. ATV-NDV tumor vaccine with attached anti-HN-anti-CD3 (αHN-αCD3) and anti-HN-anti-CD28 (αHN-αCD28) bsAb exerted in vitro, upon stimulation of allogeneic human peripheral blood mononuclear cells (PBMC), strong and durable antitumor effects against human tumor cell monolayers [152]. This anti-tumor activity was independent from recognition of TAs.

The activity of the bsAbs could be further augmented by introducing the cytokine IL-2 which binds to its high affinity receptor CD25 on T cells. For this purpose, a trispecific immunocytokine reagent (αHN-IL-2-αCD28) was constructed, produced and tested in vitro as above [153]. A transcriptome analysis and cytokine profiling of naive T cells stimulated by ATV-NDV tumor vaccine via attached αHN-αCD3 (for signal 1) and αHN-IL2 (for signal 2) revealed unsuspected costimulatory activity of the cytokine IL-2 [154]. The above trispecific immunocytokine, attached to ATV-NDV, provided the strongest T cell costimulatory activity in combination with a suboptimal amount of attached αHN-αCD3. This effect was most likely due to the concomitant transmission of signal 1 via CD3 and costimulatory signals 2a and 2b via the two T cell co-receptors CD28 and CD25 [152]. 

A Phase I clinical study evaluated the recombinant bispecific protein αHN-αCD28. In this autologous situation, the ATV-NDV vaccine provided TAs for signal 1 and αHN-αCD28 for signal 2. This dose-escalation study, in which the vaccine was modified by increasing amounts of the bsAb, involved 14 colorectal carcinoma patients with late-stage disease (stage IV with liver metastases). There were no severe adverse events. Before the vaccination, none of the patients had detectable levels of cancer-reactive blood circulatory T cells (ELISPOT test). Interestingly, after vaccination, all patients became positive in the ELISPOT assay, at least once during the course of 5 vaccinations. Furthermore, there was a dose-response relationship with the bsAb. A partial response of metastases was documented in 4 patients. The study suggests that the bsAb modified vaccine ATV-NDV-bsαHN-αCD28 is safe and can re-activate possibly anergic T cells from advanced-stage cancer [155]. A further potentiation of the costimulatory effect can be expected from a vaccine of the type ATV-NDV-αHN-*IL-2*-αCD28. 

Bispecific antibodies and also trispecific immunocytokines have been proposed to have great potential in future for targeting the immune system against cancer [156]. While NDV infection of tumor cells could break T cell tolerance in vitro with human cells [89], it was not sufficient to have a similar effect in vivo in late-stage cancer patients. To overcome anergic T cells in this situation and to reactivate them required stronger costimulation as exemplified with the bispecific anti-CD28 antibody attached to ATV-NDV [155]. Not only the costimulatory signal can be augmented by bsAb but also signal 1. If the TA mediated signal 1 is very weak, it could be intensified by a suboptimal amount of bsαHN-αCD3. The ATV-NDV-bsαHN-αCD28 vaccine would be further modified by attachment of bsαHN-αCD3.

The concept of increasing T cell costimulatory signals is complementary to the concept of decreasing T cell inhibitory signals via checkpoint inhibitory antibodies. It is likely that the ratio of positive to negative signals will be decisive for the T cell response. 

Table 3 contains a summary of the mentioned studies of clinical application of NDV.

## 7. A New Concept of Multimodal Cancer Immunotherapy

### 7.1. Treatment of Cancer Patients with Autologous Viral Oncolysate Pulsed DC Vaccine (IO-VAC^R^)

Another strategy of further improvements of the NDV-modified vaccine was to combine it with DCs. This would allow de novo generation of TA-specific T cells from naive T cells. A protocol was developed by the Immune-Oncological Center (IOZK) in Cologne (Germany) for the generation of DCs from patient-derived blood monocytes. These were then loaded with viral oncolysate (VOL) from ATV-NDV and matured by a cytokine cocktail. The product, IO-VAC^R^ (formerly called viral-oncolysate-pulsed DC, or VOL-DC), produced under Good Manufacturing Practice (GMP), received in 2015 an official production permit. This is a specific, autologous anti-tumor directed DC vaccine for intracutaneous application. This permit required, among others, the production of GMP-quality oncolytic NDV (an attenuated variant from the mesogenic strain *Mukteshwar,* also called *MTH-68*). Production of GMP-quality NDV was achieved for the first time, worldwide.

### 7.2. The Two Steps of Multimodal Immunotherapy at IOZK

IOZK developed, apart from *GMP-NDV* and the vaccine *IO-VAC^R^* also a protocol of multimodal immunotherapy, performed individually as an outpatient. The rationale behind this and further details have been described [81]. The concept of multimodal immunotherapy has recently been supported by the observation that pre-existing immunity to oncolytic virus (NDV) potentiates rather than inhibits its immunotherapeutic efficacy [114]. The multimodal immunotherapy consists basically of two steps: First, “in situ” vaccination, second, active-specific vaccination.
Immune triggering of the cancer patient’s immune system. This is done by combining systemic (i.v.) application of NDV with a physical modality inducing ICD: Local moderate electrohyperthermia (mEHT). mEHT is a type of hyperthermia. This is only one of several physical modalities inducing immunogenic cell death (ICD). Others are ionizing irradiation, ultraviolet C light, photodynamic therapy with hypericin, and high hydrostatic pressure [157]. This pre-treatment by mEHT and NDV, which takes about one hour per day, is repeated 5 times over a period of 5 days. Since both, mEHT and NDV, induce in tumor cells ICD, this first step of treatment can be considered as “in situ” vaccination [158].Active-specific vaccination is then performed at the eigth day of the treatment by intradermal application of the “ex vivo” produced vaccine IO-VAC^R^ (see above). This is done in combination with a sixth dose of NDV and sixth session of mEHT. The patient-derived DCs for the preparation of IO-VAC^R^ are obtained by a differentiation process from adherent monocytes (CD14++,CD86+,CD209−,CD83−) via semiadherent immature DCs (CD14+,CD86+,CD209++,CD83−) to floating mature DCs (CD14−,CD86++,CD209+,CD83++). This process is followed regularly by flow cytometry.

In principal, this multimodal immunotherapy activates and guides the patient’s immune system specifically toward the patient’s tumors TAs. The evoked response is individually specific and broadly polyclonal, depending on the individual tumor’s repertoire of neopeptides and common TAs. The treatment strategy takes into account the actual TAs available at all tumor locations in the body. This broad polyclonal response can be considered to be similar to the reported generation of multivirus-specific T cells by a single stimulation of peripheral blood mononuclear cells with a peptide mixture [159]. 

Among the cancer patients that have been treated at IOZK within the last 14 years are patients with more than 70 different types of cancer. For further details see www.iozk.de. Since the treatment is highly individual, the results can only be evaluated as single case or as case-series studies.

One case report demonstrates long-term remission of prostate cancer with extensive bone metastases [160]. The patient had failed standard therapy, but then achieved complete remission following multimodal immunotherapy at IOZK. Another case report relates to long-term survival of a breast cancer patient with extensive liver metastases [161]. After operation, the patient had refused systemic standard therapy and received instead the multimodal immunotherapy from IOZK. The patient survived more than 66 months after the initial diagnosis and had a continuous high quality of life. A long-lasting tumor-reactive memory T cell responsiveness was documented and served to manage the follow-up treatment.

The efficiency of multimodal immunotherapy [81] was also studied as part of first line treatment for patients with glioblastoma [162]. A retrospective analysis was performed in March 2018. It included 63 adults (median age 60 years) with first diagnosis of primary GBM in whom immunotherapy was added in conjunction with first line treatment involving maintenance therapy with Temozolomide (TMZm). Fifteen patients had chemotherapy combined with ICD treatment via NDV injections and mEHT. The median progression-free survival was 13 months. With a median follow-up of 17 months, median OS was not reached. An estimated OS at 30 months was 58%. 

An update of the data of this patient group was calculated in May 2019. The updated OS curve is shown in Figure 1. At 3 years, the OS was 50%. This compares very favorably with results obtained in an earlier ATV-NDV vaccination study. At 3 years, in that GBM clinical study, the OS in the vaccinated group was 4% while in the control arm it was 0% [140]. The OS curve of Figure 1 is reminiscent of the OS curve of colon cancer stage IV patients treated by the vaccine ATV-NDV in a randomized-controlled study [142,143]. Both study curves reveal a high plateau tail of long-term survival.

The data suggest that the additional induction of ICD via multimodal immunotherapy during TMZm cycles is beneficial in improving survival. This can be explained as follows: While TMZm targets only dividing tumor cells, ICD targets dividing and non-dividing tumor cells [163]. In addition, NDV enhances the growth-inhibiting and proapoptotic effects of TMZ on GBM cells in vitro and in vivo [107]. Of importance are also the immune stimulatory effects of NDV (Table 1) and its anti-neoplastic effects (Table 2), in particular the targeting of the oncogenic protein Rac1. 

With regard to the mechanism of the therapeutic effect, in the above mentioned GBM ATV-NDV vaccination study [140], vaccine-induced immune effects had been documented at various levels: (i) significant increase of skin delayed-type hypersensitivity tumor-reactive responses, (ii) significant increase in blood-circulatory tumor-reactive memory T cells and (iii) significant increase in CD8+ tumor-infiltrating T cells (TILs) that had infiltrated across the blood-brain barrier into secondary GBM tumors [140]. It is likely that this general increase of the number of tumor-reactive memory T cells can also explain the long-term survival effect in the GBM vaccination study with IO-VAC^R^ [162].

The generation of the vaccine IO-VAC^R^ so far required material from operated tumor samples to generate the TAs for DC loading. To broaden the application of its multimodal immunotherapy, IOZK develops at present a procedure to become independent of an operated tumor sample: in situ vaccination. 

The first step is identical to step 1 above. The second step involves the use of extracellular vesicles, such as antigenic extracellular microvesicles and apoptotic bodies as a source of TAs for DC loading. It was reported recently that oncolytic NDV uses exosomes (NDV-Ex) to transport viral proteins [56] or micro-RNA [57] into neighboring cells. Similarly, DCs produce exosomes to cross-present TAs and TLR ligands to other DCs [94]. The induction of ICD during the 5 days pre-treatment as first step of the multimodal immunotherapy thus is likely to produce ICD products including exosomes. Whether these can be used to produce an IO-VAC^R^ vaccine, depends on further research including diagnostics.

Immune diagnostics is performed at IOZK before the patient receives his/her recommendation of treatment. Immune diagnostics is also performed during follow-up. This includes the determination of Apo10 protein epitope (Apo10) and Transketolase-like 1 (TKTL1) in monocytes, the mRNA expression level for PD-L1 on circulating tumor cells, and the Th1/Th2 ratio [163]. A liquid biopsy method was developed to exploit reactivity of the innate immune system towards cancer. It is called Epitope detection in monocytes (EDIM) and tests via FACS analysis blood monocytes co-expressing Apo10 and TKTL1. Apo10 protein epitope characterizes tumor cells with abnormal apoptosis and abnormal proliferation while TKTL1 is a marker for anaerobic glucose metabolism, also known as Warburg effect. This test had a sensitivity of 95.8% and a specificity of 97.3% when evaluated for patients with prostate, breast and oral squamous cell carcinomas [164].

This EDIM/PanTum Detect test was reliable to monitor the response to multimodal immunotherapy treatment in children with brain cancer. This is a rare disease worldwide. The tumor is mostly not operable and there is no therapy available. Seventy-six children (median age 6.5 to 9.5 years) were treated at IOZK with the multimodal immunotherapy to induce ICD, in situ vaccination and postoperative anti-tumor immunity. Since no primary tumor material was available, an innovative procedure was used to prepare the IO-VAC^R^ vaccine: After 5 days of pre-treatment with mEHT plus NDV infusion, the EDIM/PanTum Detect assay revealed a significant increase in the blood of monocytes that had phagocytosed tumor material, as revealed by Apo10 and TKTL1 positivity. Therefore, at day 5 of treatment, serum was taken from the blood of the children as a source of antigenic extracellular microvesicles and apoptotic bodies to load their DCs. 

This new treatment was feasible and safe. There were no intervention-related side effects. Median OS of 28 children with diffuse intrinsic pontine glioma (DIPG) was 14 months, with a 2-year OS of 12%. Multimodal immunotherapy in children with brain tumors contributed to improved tumor control and overall survival. 

If the new concept of in situ vaccination [158] comes through the developmental steps, it could become a change of paradigm. There would be an off-the-shelf protocol to generate an immune response as close as possible to the patient’s tumor as it exists at the time of treatment. Since tumors are heterogeneous and change during time, such a protocol would enable to go along with such changes. The applications would be broad and yet individual and even tumor time-course adapted.

Table 4 contains a summary of clinical application of NDV via multimodal immunotherapy.

Table 5 lists the types of cancer with reported sensitivity to NDV virotherapy. The quoted references relate to this review. It is of special significance that this list includes the most frequent types of cancer in human, the carcinomas: those from breast, ovaries, colorectum, pancreas, kidney, prostate, lung, and liver. Tumors from other types of tissue were also found to be sensitive to NDV: melanoma, glioblastoma and lymphoma.

### 7.3. Side Effects

A recent review compares side effects of chemotherapy to those of biological therapy and describes novel concepts to reduce side effects from systemic cancer treatment [165]. Side effects of grade 1–4 are reported for cytostatic drugs, small molecule inhibitors (SMIs), and checkpoint inhibitory mAbs. Side effects of grade 1–3 are reported from chimeric antigen-receptor (CAR) T cells and anti-tumor mAbs. In contrast, therapy with cancer vaccines and with OVs like NDV has only low side effects (grade 0–2). 

Further details about side effects of standard therapy (surgery, radiotherapy, chemotherapy, hormone therapy), biological therapies, targeted therapies with small molecule inhibitors, immunotherapy, oncolytic virotherapy and combination therapies can be found in a recent book entitled “Quo vadis cancer therapy?” [166].

### 7.4. Limitations and Challenges

The review has outlined the great potential of NDV for breaking therapy resistance and for improving cancer therapy. However, there is still a long way to go. There exists no optimized standard protocol for the mode and dose of virus application to patients. All the studies mentioned are experimental. There is a conflict between clinical efficacy and the rules for performing clinical studies. For example, the remarkable results from Figure 1 have been obtained with a therapy that is highly individual and multimodal. The challenge is to adapt the rules for clinical studies to the needs of immunotherapy and of efficacy.

## 8. Future Perspective: OV-mediated Gene Therapy and Combinatorial Approaches with Bispecific Antibodies and Checkpoint Inhibitors

### 8.1. OV-Mediated Gene Therapy 

An advantage of incorporating a therapeutic gene into an oncolytic virus (OV) is that the therapeutic gene will be amplified selectively in tumor cells. “Gene therapy of cancer” has been the title of a 700-page book from 2009 including methods and protocols in molecular biology [167]. It contains also a chapter on NDV as a promising vector for viral therapy, immune therapy, and gene therapy of cancer. This topic was later further analyzed in more detail [168].

A milestone in OV-mediated gene therapy was in 2015 the approval of T-VEC (*talimogene laherparepvec*) for melanoma immunotherapy. It took about 25 years to develop this reagent [1]. In 1991, *herpes simplex virus* was genetically engineered to generate a mutant with reduced neurotoxicity. In 1997, an albumin promoter/enhancer was introduced for targeting hepatoma. In 2001, HSV was modified with transgenes encoding IL-12 and GM-CSF to improve T cell recruitment and immune stimulation [1]. The first clinical study with intralesional application of T-VEC began in 2009 [169]. 

Other OV-based gene therapy vectors were developed, among others, from *adenovirus*, *measles virus*, *vaccinia virus,* and NDV. A major aim was to improve OV therapy by overcoming barriers which exist in the blood and also within a solid tumor mass. Such barriers include (i) extracellular matrix (ECM), (ii) interstitial tissue pressure, and (iii) host innate or acquired immune effects.

The development of NDV as a vector for expressing a foreign gene was facilitated through introduction of an internal ribosomal entry site [170,171]. This provided direct proof for a sequential transcription mechanism. The strategies to produce vectors for NDV-mediated gene therapy can be categorized as follows: 

(i) Improvements in specificity by targeting TAs. One strategy was to engineer the F protein of NDV to be cleavable exclusively by prostate-specific antigen (PSA) [172]. PSA-retargeted NDV efficiently lysed prostasphere tumor mimics, suggesting efficacy in vivo. Another TA that was targeted by recombinant NDV (rNDV) was CD147, expressed by hepatocellular carcinoma (HCC). A mouse-human chimeric antibody (cHAb18) was cloned into NDV to generate the recombinant virus rNDV-18HL. The vector replicated selectively in orthotopic HCC xenografts leading to cHAb18 antibody expression in situ. This combination therapy induced tumor necrosis, reduced the intrahepatic metastases, and prolonged the survival in mice [173].

(ii) Improvements in potency. This can be achieved by introducing immune stimulatory cytokines (e.g., GM-CSF [174], IL-2 [175], IL-15 [176], costimulatory ligand ICOSL [177], or by targeting apoptosis pathways via TRAIL [178], Fas [179], apoptin [180], or p53 [181].

(iii) Improvements in delivery and spread. One concept was to increase the activity of the fusion protein and to introduce the interferon antagonist NS1 [182]. Another concept combined vascular normalization with oncolytic NDV immunotherapy in a pre-clinical model of advanced-stage ovarian cancer [183].

Of special interest is the generation of rNDV expressing a full IgG antibody from two transgenes [184]. Transgenic expression of the heavy and light chains of a monoclonal antibody (mAb), as separate additional transcriptional cassettes, led to expression of an intact IgG mAb targeting tumor angiogenesis. Infection of tumor cells with antibody-transgenic viruses resulted in efficient production and secretion of a functional mAb that specifically bound to its target antigen in tumor tissue. Similar effects were achieved by rNDV expressing a chimeric antibody against HCC [173]. OVs can also be equipped with genes for BITEs, which cross-link cancer cell’s TAs with T cell’s CD3. By such co-engagements, BITEs can mediate immune- mediated tumor cell lysis [185].

Also recently, the CRISPR-Cas9 system has been proposed as a powerful tool for efficient creation of rOVs with therapeutic genes [186].

Experts from Canda, Germany and the USA contributed to a volume of Frontiers in oncology and Frontiers in immunology entitled “Harnessing oncolytic virus-mediated antitumor immunity” [1]. There was a remarkable accord among the experts that the process of viral oncolysis [116] was considered of less importance than the induction of post-oncolytic systemic T cell mediated immunity. 

### 8.2. Combining NDV with Carrier Cells for Improving Tumor Targeting

Several groups have explored carrier cell systems for improving systemic delivery of viruses. Mesenchymal stem cell (MSC) carriers protected oncolytic *measles viruses* from antibody neutralization in an orthotopic ovarian cancer therapy model [187]. Human mesenchymal stromal cells delivered systemic oncolytic *Measles* virus to treat acute lymphoblastic leukemia in the presence of humoral immunity [188]. 

MSC enhanced the oncolytic effect of NDV in GBM and glioma stem cells (GSC) via the secretion of TRAIL [189]. NDV induced a dose-dependent cell death in GBM cells and a low level of apoptosis and inhibition of self-renewal in GSC cells [190]. Similarly, NDV was shown to bind to and be released from activated T cells [191]. In a tumor neutralization assay in vitro, monolayers of human tumor cells could be completely destroyed by the addition of polyclonally activated T cells loaded with oncolytic NDV (strain *Italien*). In this process, synergistic effects between CTL activity and viral oncolytic activity were apparent in the tumor contact zone [190]. Thus, it can be anticipated that in vivo hitchhiking of NDV on MSC or immune T cells will improve tumor targeting and therapeutic effects of NDV.

### 8.3. Combinatorial Approaches of OVs with Bi- or Tri-specific Antibodies

Specific tumor targeting of NDV was described by means of a bi-specific fusion protein [192]. This consisted of an anti-HN antibody scFv linked to the gene for IL-2 (αHN-IL-2). In vitro, retargeted virus-binding to IL-2R (CD25) positive tumor cells allowed the transfer of the marker gene enhanced green fluorescent protein (EGFP) from the recombinant NDFL-EGFP virus [192]. In vivo, biodistribution studies and quantitative real-time RT-PCR for EGFP mRNA revealed that transgene delivery was reduced by 35–100% in liver, spleen, kidney, lung and thymus by the αHN-IL-2 modified virus while 98% of the transgene was delivered to IL-2R+ tumors [191]. The bispecific protein greatly reduced unwanted side effects of systemically applied NDV in normal tissues [193]. Tumor therapy effects were mostly undiminished [193]. There is potential for augmentation of this effect by including in the virus one or more therapeutic genes.

Targeting an introduced viral antigen such as HN from NDV in tumor tissue by grafted T cells and DCs has been proposed to be possible via cell-bound tri-specific antibodies, such as αHN-IL-2-αCD28 [194]. This idea, patented in the US and in Europe, consists of 5 steps. 1. Pre-conditioning of the tumor microenvironment in the patient, for instance by mEHT (see above), 2. Local or systemic application of oncolytic NDV for introduction of HN, 3. Activation of the patients T cells and loading with tri-specific antibodies thus exposing multiple anti-HN binding sites, 4. Grafting the T cells back to the pre-conditioned patient [195].

### 8.4. Combinatorial Approaches with Checkpoint Inhibitors

Recognition of tumor immune evasive mechanisms led to the development of immunotherapeutic antibodies targeting co-stimulatory and co-inhibitory receptors on T cells. The goal consists of enhancement of T cell activation or reversal of tumor-induced T cell inhibition. Checkpoint inhibitory antibodies such as αCTLA-4 or αPD-1 have demonstrated significant promise in clinical trials, at least for a subset of patients. Combination strategies thus aim to extend therapeutic benefit to a broader range of patients. OVs have recently been recommended quite strongly to serve such a purpose. They have been proposed as antigen-agnostic cancer vaccines [196] and to make tumors “hot” [194]. 

OVs are one modality by which the effect of checkpoint inhibitors can be augmented, as shown by pre-clinical studies involving NDV. Zamarin and colleagues demonstrated that localized virotherapy with NDV induced PD-L1 in the tumor microenvironment [197] and that targeting of PD-L1 was able to break resistance to OV therapy. Similar effects have been described in studies employing *measles virus* [198], *vaccinia virus* [199] *vesicular stomatitis virus* [200], and *sindbis virus* [201]. A key role of the type I IFN pathway in OV therapy was also highlighted [202].

A further innovation consists of the use of a prime-boost schedule using separate OVs in combination with checkpoint blockade [203]. In a subcutaneous B16 melanoma mouse model, intra-tumoral inoculation of *Reovirus* primed CD8+ T cells and CD4+ Th1 T cells. Boosting was performed with VSV expressing a cDNA library of melanoma antigens (VSV-ASMEL). Systemic delivery promoted a potent anti-tumor CD4+ Th17 response. The combination with anti-PD-1 significantly enhanced survival, including long-term cures [202]. 

In a window-of-opportunity clinical study, it was recently shown that intravenous delivery of oncolytic *Reovirus* to brain tumor patients immunologically primed for subsequent checkpoint blockade [204]. Furthermore, neoadjuvant oncolytic virotherapy before surgery has been demonstrated to sensitize triple-negative breast cancer in a therapeutic model to immune checkpoint blockade [205].

Of importance for future improvements of the effectivity of checkpoint blockade immunotherapy is a detailed understanding of the tumor microenvironment. The prognostic impact of tumor-infiltrating cells depends on their localization (e.g., invasive front or tumor center), on the presence of tumor-adjacent lymphoid aggregates and on the type of inflammatory context [203].

Stromally located tumor-infiltrating lymphocytes (sTILs) provided a good prognosis in early-stage triple-negative breast cancers [206]. The tumor microenvironment, however, presents physical, immunological and metabolic barriers to durable immunotherapy responses. Examples of the latter are metabolic insufficiency and tumor hypoxia [207]. TGFβ was recently reported to attenuate tumor response to PD-L1 blockade by contributing to exclusion of T cells [208]. Successful anti-PD-1 cancer immunotherapy required T cell-DC crosstalk with a licensing loop involving the cytokines IFN-γ and IL-12 [209]. Furthermore, a single transcription factor, TCF7, was visualized within CD8+ T cells in fixed melanoma tumor samples and predicted positive clinical outcome to checkpoint immunotherapy [210]. 

OVs are well suited to convert “cold”, non-infiltrated, tumor areas into “hot” ones that can become infiltrated. Other modalities include radiation, chemotherapy, small molecule inhibitors, histone deacetylase (HDAC) inhibitors [211], vaccines, TLR agonists, or type I interferon. The search is also directed towards other immune checkpoint inhibitors (e.g., TIM-3, LAG-3, TIGIT, and BTLA) or agonists of T cell costimulatory receptors (e.g., 4-1BB, OX40, GITR) [212]. Other combinatorial agents might target inhibitory mechanisms such as indoleamine dioxygenase (IDO), Tregs and myeloid-derived suppressor cells (MDSCs) [213,214].

A Phase 1b clinical trial with advanced melanoma (*n* = 21) confirmed the synergistic effect between oncolytic virotherapy (applying T-VEC) and anti-PD-1 immunotherapy. The promoted intratumoral T cell infiltration observed after OV therapy suggested that the improved efficacy of anti-PD-1 therapy was due to a change in the tumor microenvironment [215].

A recent report provides an example of a triple combination strategy [216]. T-VEC was combined with MEK inhibitors which increased viral replication independent of mutation status. Combination therapy in murine melanoma increased PD-1/PD-L1 expression and PD-1 blockade further enhanced tumor regression. Combining small molecule inhibitors, OVs, and immune checkpoint blockade was proposed as a triple threat to cancer [217].

### 8.5. Modulation of OV Therapy by Pharmacologicals

Pharmacological modulation of anti-tumor immunity induced by oncolytic viruses has been reviewed in an article by NE Forbes and colleagues in a special issue of Frontiers in Oncology [218]. Here, only some main examples will be given:Combining OVs with DNA alkylating drugs. Two examples are cyclophosphamide (CPA) and Temozolomide (TMZ). CPA has been used successfully in combination with OVs including *HSV-1, adenovirus, vaccinia virus, reovirus, measles, and vesicular stomatitis virus.* The best progression-free survival and OS rates were seen with a combination of low-dose metronomic CPA and intratumoral infection by gene-modified adenovirus Ad-GM-CSF [216]. In another study, using Ad5/3-D24-GM-CSF with or without low-dose CPA to reduce Tregs, co-treatment with TMZ increased tumor cell autophagy, anti-tumor immunity, and reduced tumor burden [219].Combining OVs with inhibition of DNA replication. Gemcitabine, a fluorinated deoxycytidine nucleoside analog, was shown to increase the anti-tumor activity of *adenovirus*, *parvovirus*, *reovirus*, *vesicular stomatitis virus*, *HSV*, *vaccinia, and myxoma virus*. A Phase I study was performed in 16 patients with advanced cancers by combining gemcitabine with intravenous *reovirus* [220].Combining OVs with epigenetic modulators. Dysfunctional IFN pathways in cancers are often due to epigenetic silencing including DNA promoter hypermethylation and transcriptionally suppressive histone modifications. Transcriptional activation of ISGs, which are often epigenetically silenced, requires HDAC activity [221]. HDAC inhibitors include valproic acid (VPA) and trichostatin A (TSA). These have been used in combination with OVs to effectively “reprogram” IFN-responsive tumors to become permissive to OV infection. Epigenetic modifiers have also been recommended to boost cancer immunotherapies [222,223]. One example will be given: 5-AZA-2′-deoxycytidine (5-AZA) is a methyltransferase inhibitor that prevents DNA methylation and allows silenced DNA to regain accessibility to transcription factors. An increase in survival was observed in glioma bearing mice treated with oncolytic HSV rQNestin34.5 and 5-AZA compared to either treatment alone [223]. 5-AZA had been found much earlier to be capable in vitro to de-repress on immunoresistant tumor variants expression of a TA which could be recognized by specific CTLs [224]. Recent advances in epigenomics have illuminated epigenetics in cancer stem cells (CSC) [225]. CSCs have been identified in a number of solid tumors, including breast cancer, brain tumors, lung cancer, colon cancer, and melanoma. Lung cancer stem cells (CSC) were recently reported to be responsible for tumor initiation, treatment resistance and tumor recurrence [225].Combining OVs with pharmacological modulation of autophagy. Chloroquine is an autophagy inhibitor and rapamycin an inducer of autophagy. Oncolytic effects of NDV (strain *FMW*) on cisplatin or paclitaxel resistant lung carcinoma cell lines could be significantly potentiated in vitro and in vivo by combination treatment with these autophagy modulators [105]. Oncolytic NDV was demonstrated to replicate in CSC-enriched lung cancer spheroids, to lyse them and to inhibit their 3D growth potential. Treatment of such spheroids with chloroquine increased the NDV-induced cytotoxicity [226].

Table 6 contains a summary of future concepts and perspectives of OV-mediated therapy.

## 9. Summary

The review has updated latest information on NDV. This oncolytic virus behaves in cancer patients as a biological agent with the capacity of tumor selective virus replication and tumor selective oncolysis. Immunogenic tumor cell death leads to induction of post-oncolytic systemic anti-tumor immunity. 

The main topic relates to breaking therapy resistance. NDV is reported to have the potential to break (i) T cell tolerance, (ii) resistance to chemotherapy or radiotherapy using X-rays and gamma rays, (iii) resistance to apoptosis, (iv) resistance to hypoxia, (v) resistance to TRAIL, (vi) resistance to immune checkpoint blockade and, as discovered recently, (vii) resistance to anti-viral immunity.

Many of its immune stimulatory properties are also described. They have to do with the induction of a strong type I interferon response. This creates an inflammatory environment that facilitates recruitment and activation of immune cells. NDV also directly activates NK cells, monocytes, macrophages and dendritic cells and polarizes the latter towards DC1. On T cells, NDV exerts costimulatory effects which are important to prevent T cell tolerance towards encounter with tumor antigens. The 22 anti-neoplastic and immune-stimulatory effects described make NDV to one of the most promising biological agents in cancer therapy. In addition, in contrast to chemotherapy or radiotherapy, oncolytic NDV therapy has almost no side effects.

Of significance is the identification of the oncogenic protein Rac1 with which NDV interacts upon viral cell entry and which is essential for viral replication. Rac1 is also reported to be important for GBM cell migration and invasion. By targeting Rac1, NDV has the potential to break therapy resistance of this deadly disease. The review shows an updated remarkable OS survival curve of 63 GBM patients treated by multimodal immunotherapy in combination with standard therapy. The side effects of the multimodal immunotherapy are very low (Grade 0–2). This is in contrast to immunotherapies such as checkpoint inhibitory antibodies, CAR T cells or some other OVs (Grade 1–3/4).

NDV is an oncolytic virus with a very long history of clinical application and with a very high safety profile. The review includes examples of breaking therapy resistance via NDV application in patients who did not respond to standard therapy. It also includes future perspectives and points towards OV-mediated cancer gene therapy and combinatorial approaches, in particular those with checkpoint inhibitors. Other promising improvements can be expected from combining OVs and immune cells with bi- or tri-specific antibodies to improve tumor targeting.

## 10. Conclusions

Based on the unique anti-neoplastic properties of NDV and on its positive interaction with the patient’s immune system, this biological agent can be considered as a major breakthrough in cancer therapy: where conventional therapies end and fail because of cancer resistances, this oncolytic virus starts to develop its full potential for improvements of cancer therapy.

## Figures and Tables

**Figure 1 biomedicines-07-00066-f001:**
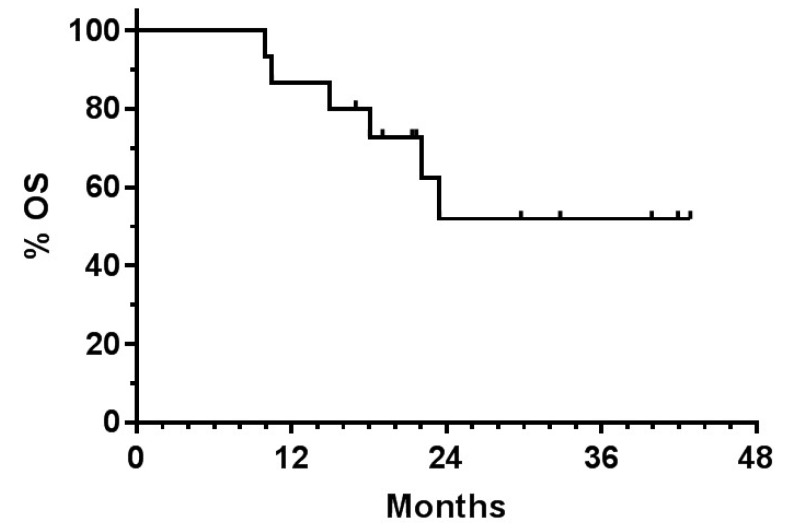
Patients with first diagnosis of primary GBM were treated with Standard therapy including multimodal immunotherapy. The details of the patients have been published [142]. An updated overall survival curve of this cohort of patients, 14 months later, is presented. The difference to the earlier calculation is due to the later time point of analysis.

**Table 1 biomedicines-07-00066-t001:** Immune stimulatory effects of Newcastle disease virus (NDV) in non-permissive hosts.

Induction of post-oncolytic immunity
Activation of natural killer (NK) cells
Activation of monocytes and macrophages
Activation of dendritic cells
Polarization of dendritic cells (DCs) towards DC1
Induction of a strong type I interferon response
IFN-α mediating epitope cross-presentation by DCs
IFN-α mediating a link between innate and adaptive immunity
IFN-α being essential for the generation of cytotoxic T cells (CTLs)
IFN-α protecting T cells against attack by NK cells
NDV exerting co-stimulatory effects on CD4+ and CD8+ T cells

**Table 2 biomedicines-07-00066-t002:** Anti-neoplastic effects of NDV in non-permissive hosts.

Targeting the oncogenic protein Rac1
Tumor-selective virus replication
Tumor-selective oncolysis
Tumor-selective induction of immunogenic cell death
Potential to break T cell tolerance towards tumor-associated antigen (TA) expressing tumor cells
Potential to break resistance to chemotherapy or radiotherapy
Potential to break resistance to apoptosis
Potential to break resistance to hypoxia
Potential to break resistance to TNF-related apoptosis-inducing ligand (TRAIL)
Potential to break resistance to immune checkpoint blockade
Potential to break resistance to anti-viral immunity
Promotion of virus propagation via syncytia, autophagy, and exosomes

**Table 3 biomedicines-07-00066-t003:** Clinical application of NDV (Part I).

1. Post-operative application of NDV oncolysate vaccines in Stage II melanoma (*n* = 83) by WA Cassel and DR Murray (1977) (oncolytic strain 73 T)
2. Systemic treatment of advanced chemorefractory cancers; Phase II placebo-controlled trial (*n* = 33 versus 26 controls) by LK Csatary and S Eckhardt (1993) (oncolytic strain MTH/68)
3. Systemic treatment of glioblastoma multiforme (GBM) (*n* = 14); Intra-patient dose escalation followed by 3 cycles of 55 billion infectious units by AI Freeman and Z Zakay-Rones (2006) (lentogenic strain HUJ)
4. Phase I dose-escalation trials of intravenous virus administration in patients with advanced solid cancers resistant to standard therapy (*n* = 113) by AL Pecora and RM Lorence (2002, 2003) (oncolytic strain PV71)
5. Post-operative treatment with the irradiated live tumor cell vaccine Autologous Live NDV-Modified Tumor Cell Vaccine (ATV-NDV): (i) Early breast cancer (*n* = 63), metastatic breast cancer (*n* = 27) and metastatic ovarian cancer (*n* = 31) by T Ahlert and V Schirrmacher (1997) (lentogenic strain Ulster) (ii) Phase II trial in patients with locally advanced colorectal carcinoma (CRC) (*n* = 57) by D Ockert and V Schirrmacher (1996) (lentogenic strain Ulster) (iii) Phase II trial in patients with GBM (*n* = 23 versus 87 non-vaccinated patients from the same clinic in the same time interval) by HH Steiner and C Herold-Mende (2004) (lentogenic strain Ulster) (iv) Phase I/II trial patients with Stage III and IV head and neck squamous cell carcinomas (HNSCC) (*n* = 20) by J Karcher and G Dyckhoff (2004) (lentogenic strain Ulster)
6. Prospectively randomized Phase II/III trial to investigate the efficiency of ATV-NDV vaccination after liver resection for hepatic metastases of CRC as a tertiary prevention method (*n* = 51) by T Schulze and PM Schlag (2009) (lentogenic strain Ulster)

**Table 4 biomedicines-07-00066-t004:** Clinical application of NDV (Part II): Multimodal cancer immunotherapy combining hyperthermia/NDV pretreatment with active-specific vaccination via IO-VAC^R^.

**Single-case Report of Prostate Cancer with Extensive Bone Metastases**
Single-case report of breast cancer with extensive liver metastases
Retrospective analysis of adults with GBM (median age 60) (*n* = 63)
Retrospective analysis of children (median age 6.5–9.5) (*n* = 76) with brain cancer; among them diffuse intrinsic pontine glioma (DIPG) (*n* = 28) 2015: IOZK succeeded in producing oncolytic NDV (derivative from strain Mukteshwar) from cell culture at high titers and with high purity according to GMP guidelines and received a permit for its use in cancer patients. This is used to produce the autologous NDV viral oncolysate-pulsed DC vaccine IO-VAC^R^.

**Table 5 biomedicines-07-00066-t005:** Types of cancer with reported sensitivity to NDV virotherapy.

Melanoma (6, 88, 108, 132, 133)
Breast carcinoma (51, 63, 69, 101, 138, 163)
Ovarian carcinoma (119, 160, 164)
Colorectal carcinoma (67, 113, 139, 142, 143, 157)
Head and Neck squamous cell carcinoma (141)
Gastrointestinal carcinoma (144)
Pancreatic carcinoma (41, 44, 45, 46)
Renal cell carcinoma (109, 134)
Prostate carcinoma (162, 174)
Lung carcinoma (104, 105, 229)
Hepatocellular carcinoma (175, 183, 184)
Glioblastoma multiforme (65, 70, 106, 140, 145, 147, 164, 191)
Lymphoma (84, 86, 87, 135, 136, 193, 194, 195, 226)

**Table 6 biomedicines-07-00066-t006:** Future perspectives.

OV-mediated gene therapy, incorporation of therapeutic genes
Improving tumor targeting of OVs by carrier cells
Improving T-cell costimulation by bi- or tri-specific antibodies
Improving OV targeting of immune cells by loading the cells with tri-specific antibodies
Combining OV therapy with checkpoint inhibitors
Combining OV therapy with physical modalities such as mEHT
Combining OV therapy with pharmacological modulation

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
