# Peer review of "Breaking Therapy Resistance: An Update on Oncolytic Newcastle Disease Virus for Improvements of Cancer Therapy"

_biomedicines, 2019, doi:10.3390/biomedicines7030066_

Round 1

Reviewer 1 Report

The authors have produced a review that updates information on oncolytic immunovirotherapy using NDV and added clinical data obtained mainly from IOZK. Various treatments including  hyperthermia, dendritic cell vaccine, and NDV administration were performed. The clinical results in the primary GBM are remarkable. These recent developments are worthy of publication.

There are some comments to clarify the authors' s intentions.

Page 10, lines 452-453, "Non-proliferating cells such as cancer stem cells or dormant tumor cells are not affected by CT or RT": This is true for conventional photon radiotherapy using X-rays and gamma rays. However, carbon ion radiotherapy is known to be effective for cancer stem cells under hypoxic condition. This needs  to be taken into consideration.   Page 16, lines 694-695, "At 3 years, in that GBM clinical study, the OS in the vaccinated group was 4% while in the control arm it was 0% (122)": In ref. 122, Schulze at al. examined colorectal cancer with liver metastasis. This discrepancy should be explained. Page 17, Figure 1 legend, "The details of the patients have been published (138)": In ref. 138, Hammerich et al. reviewed in situ vaccination. It needs to be rectified. Page 17, Figure 1: In ref. 142, Van Gool et al. indicated that "estimated median overall survival at 30m was 58%, whereas in Figure 1, the OS was 50%. This discrepancy should be explained. Page 14, table 3: In there a difference between "post-surgical" and "post-operative"? References: The page notation style needs to be reconfirmed.

Author Response

Thank you for the fast review. Here are the revisions: 1. Page 10 has been modified, 2. Page 16 has been corrected, 3. Page 17, the figure legend has been extended, 4. Paper to in situ vaccination has been quoted (now Ref 160), 5. The two evaluations were performed at different time points. At the earlier time point the survival was 58%, 14 months later it decreased to 50%. This is not unexpected. The survival is still very high. 6. Page 14, post-surgical was the term from the author and included primary tumor and regional lympg nodes. It has now been changed to postoperative, 7. Page notation has been adapted. 

Reviewer 2 Report

The manuscript entitled "Breaking therapy resistance: An update on oncolytic Newcastle disease virus for improvements of cancer therapy" by Volker Schirrmacher, Stefaan Van Gool, and Wilfried Stuecker describes the history and recent progress in the use of NDV as an oncolytic agent indicating the ability of NDV to break therapy resistance and underlines the antiviral immunity aspects in tumor treatment. The work is outstanding due to very comprehensive and detailed description of the topic.

There are few minor improvements needed:

1) Please provide the Latin nomenclature for the virus families (lines 40-42). 2) Please unify the nomenclature of the viruses mentioned throughout the manuscript. Capital letters (Respiratory Syncytial virus), etc. are not used in the nomenclature, like for viruses with geographic names. Otherwise, we simply write respiratory syncytial virus. Also, for the name paramyxovirus (line 122) we do not use capital letters. 3) Lines 372, 378: some letters in pdf file are missing (IFN- ). 4) Lines 573-574: please verify this information and provide proper units (PFU/m2).

Overall, this manuscript deserves to be published after minor revision.

Author Response

We thank for the positive evaluation. Here is our response: 1. We prefer the english nomenlature for the virus families, as we see it in many other reviews, 2. Capital letters in virus nomenclature has been changed, 3. greek letters have been corrected, 4. PFU/m2 has been included (line 695), in Ref 147 such information had not been given.

Reviewer 3 Report

The manuscript submitted by Schirrmacher et al. entitled "Breaking therapy resistance: An update on oncolytic Newcastle disease virus for improvements of cancer therapy" presents a comprehensive review on NDV focusing on molecular biology, anti-neoplastic and immune-stimulating effects, tumor immunity, clinical experience with NDV. Although, the manuscript is well written and structured few points are recommended for further consideration:

1) Indeed, NDV has been used in a number of clinical trials. For instance, NDV expressing multiple tumor-associated antigens (TAAs) has been demonstrated to provide long-term survival in phase II trials in patients with ovarian, stomach, and pancreatic cancer. In turn, in melanoma patients immunized with NDV in a randomized double-blind phase II/III trial the study results suggested that there were no remarkable differences between the vaccinated individuals and those in the placebo group. Also It was demonstrated that vaccination with NDV provided prolonged survival and short-term improved quality of life (Kenneth Lundstrom, Diseases, 2018, 6(2): 42 DOI: 10.3390/diseases6020042).

Authors provide arguments and rationale for the application of NDV for cancer therapies. However, what is missing is a description of the virus limitations, challenges e.g. with clinical studies, efficacy.

2) Would be interesting and educative for the potential readers to compare NDV to other OVs in terms of their immunogenecity, anti-cancer and immunomoduatory properties especially with those already in a clinic (T-VEC) or with promising clinical data (e.g. adenoviruses, coxackievirus).

3) Suggest to provide more latest references to other OVs (e.g. adenoviruses) - up to date reported findings (pre-clinical and clinical), future prospects.

4) The manuscript should be review by the English native speaker.

Author Response

We thank the reviewer for the productive comments. Here is our response: 1. We did not quote the melanoma study by Voit C et al. mentioned in the review by K Lundstrom. It is a poor study. It employed a lysate vaccine instead of a live cell vaccine. In the quoted paper by Ahlert et al. (138) it was demonstrated that a lysate vaccine is of low quality. In addition, a randomized study comparing 9 versus 8 patients is not very meaningful, 2. We have introduced a paragraph on limitations and challenges (lines 918-926), 3. We introduced a chapter comparing NDV to the mentioned other OVs (line 522-624), 4. more references to other OVs have been included (e.g. Ref 115-130).